# Hierarchical Object-Aware Dual-Level Contrastive Learning for Domain Generalized Stereo Matching

**Yikun Miao**
Beijing Institute of Technology
joshmiao233@gmail.com

**Meiqing Wu** *
Nanyang Technological University
meiqingwu@ntu.edu.sg

**Siew-Kei Lam**
Nanyang Technological University
assklam@ntu.edu.sg

**Changsheng Li**
Beijing Institute of Technology
lcs@bit.edu.cn

**Thambipillai Srikanthan**
Nanyang Technological University
astsrikan@ntu.edu.sg

## Abstract

Stereo matching algorithms that leverage end-to-end convolutional neural networks have recently demonstrated notable advancements in performance. However, a common issue is their susceptibility to domain shifts, hindering their ability in generalizing to diverse, unseen realistic domains. We argue that existing stereo matching networks overlook the importance of extracting semantically and structurally meaningful features. To address this gap, we propose an effective hierarchical object-aware dual-level contrastive learning (HODC) framework for domain generalized stereo matching. Our framework guides the model in extracting features that support semantically and structurally driven matching by segmenting objects at different scales and enhances correspondence between intra- and inter-scale regions from the left feature map to the right using dual-level contrastive loss. HODC can be integrated with existing stereo matching models in the training stage, requiring no modifications to the architecture. Remarkably, using only synthetic datasets for training, HODC achieves state-of-the-art generalization performance with various existing stereo matching network architectures, across multiple realistic datasets.

## 1 Introduction

Stereo matching aims to find horizontal pixel-wise displacement, *i.e.*disparity, between a rectified stereo image pair to recover depth for applications including autonomous driving, robotics, and augmented reality. In recent years, deep learning based stereo matching networks have achieved state-of-the-art performance in multiple benchmarks [12, 28, 32, 33], benefiting from the expressive power of deep feature representations.

A typical stereo matching pipeline [3, 13, 51, 34] includes four stages: feature extraction, cost volume generation, cost aggregation, and disparity regression, which is usually trained in an end-to-end way, supervised by ground-truth disparity. As collecting dense annotations of disparity for real-world datasets is often costly, large-scale synthetic datasets (*e.g.*, SceneFlow [27]) are widely adopted to train stereo matching networks. This however often leads to failures while generalizing to unseen

---

*Corresponding Author

38th Conference on Neural Information Processing Systems (NeurIPS 2024).

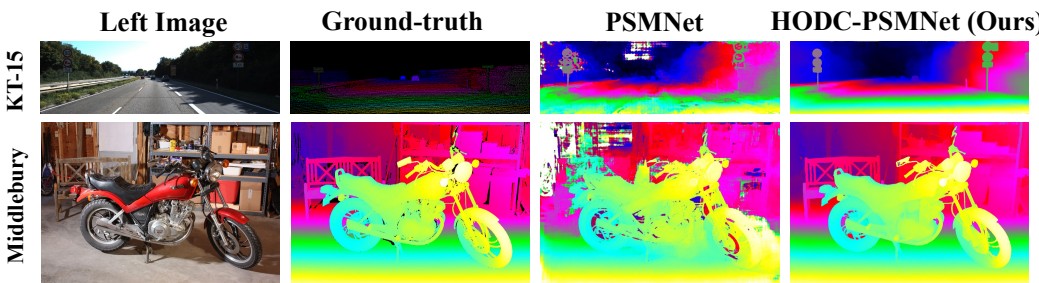

Figure 1: Qualitative domain generalization results of PSMNet [3] baseline and HODC-PSMNet (Ours). The latter is trained with our proposed hierarchical object-aware contrastive loss. Both models are trained only on the *synthetic* SceneFlow [27] dataset and evaluated directly on *realistic* datasets KITTI-2015 [28] and Middlebury [32].

realistic domains (see the third column in Fig. 1 for example), as stereo matching networks will over-rely on superficial cues (*e.g.*local chromatic attributes like color distribution, illumination, texture, *etc.*) within synthetic data for disparity estimation.

Recently, several works [7, 49, 31, 4, 48, 34, 35, 16] have attempted to address the synthetic-to-real generalization gap for stereo matching networks. In order to mitigate over-reliance on the short-cut features, they either add additional regularization to enforce the feature consistency between left and right views [49], original images and adversarial perturbations [7], augmented transformations [4], or introduce specially designed network architectures to enlarge the reception field and extract robust structural and geometric representations [48, 34, 35, 16]. Though these approaches have shown promising results in their generalization ability, they do not exploit semantic structure directly as they depend mainly on pixel-wise loss to optimize disparity predictions.

We claim that semantic structure serves as an important cue to assist correspondence matching, especially in ambiguous areas like textureless regions and edge boundaries. For example, the disparity estimation for road and street signs in the first row, and motorcycle wheels in the second row in Fig. 1 are intuitively erroneous, as they violate the semantic consistency prior wherein disparity should have similar distribution within a local region on the same object. It is however non-trivial to incorporate semantic information to guide the stereo matching effectively. Early efforts [44, 36] have explored the semantic and structural information by introducing sub-networks for semantic segmentation [44] or edge detection [36] in a multi-task learning manner. However, multi-task joint feature learning couples the two different tasks superficially and cannot incorporate semantic structure to directly guide stereo matching at the ambiguous areas [26].

In this paper, we propose a novel method to effectively enhance semantically and structurally driven matching to boost the synthetic-to-real generalization capability of stereo matching networks. Inspired by the recent success of contrastive learning in learning distinctive visual representations [14, 5, 6, 39, 40, 49, 22, 38, 1, 10, 30, 15, 17, 50, 8], we introduce hierarchical object-aware dual-level contrastive learning (HODC) to incorporate the semantic guidance into the stereo matching process. Given the left and right images with object index and ground-truth disparity, we first segment the object index map at different scales to obtain hierarchical object-aware regional representations. Then, guided by the ground-truth disparity, we establish accurate correspondences between the representations of the stereo pair. Using dual-level contrastive learning, we enhance both intra- and inter-scale matching of the hierarchical object-aware regional representations from the left image to the right. Building correspondence between these representations enables us to introduce semantically and structurally driven matching in an explicit and effective manner. As illustrated in Fig. 2, the proposed method can be easily plugged into different existing stereo matching pipelines without requiring modifications to their architectures. Extensive evaluations show that the proposed method can substantially improve the synthetic-to-real generalization ability of different stereo matching networks across most stereo matching benchmarks and outperforms existing domain generalized methods by a large margin.

In summary, our main contribution is threefold:

- We demonstrate that semantically and structurally driven matching is crucial for the domain generalization ability of stereo matching networks. To achieve this, we propose a novel

hierarchical segmentation scheme on objects, enabling the extraction of hierarchical object-aware representations in a flexible manner.

- We effectively enhance the matching between hierarchical object-aware representations from the left image to the right image by introducing intra- and inter-scale dual-level contrastive loss, enabling semantically and structurally driven matching both locally and globally.

- Extensive experiments conducted on four widely used realistic stereo matching datasets using multiple network architectures underscore the effectiveness and intrinsic generality of HODC in finding semantically and structurally driven matching for generalizable stereo matching networks.

## 2 Related Work

### 2.1 Learning Based Stereo Matching Networks

In the past decade, deep learning has made remarkable strides in computer vision. In stereo matching, Zbontar *et al.*first introduced convolutional neural networks (CNNs) to extract features and compute matching cost [46]. Mayer *et al.*proposed the first end-to-end correlation-based stereo matching network DispNetC [27]. SegStereo [44] and EdgeStereo [36] incorporated semantic and edge information to resolve ambiguities in stereo matching.

More recently, several end-to-end stereo matching architectures have emerged. GCNet [19] concatenated left and right features directly to form a 4D cost volume and exploit 3D CNNs to aggregate matching costs. PSMNet [3] proposed the spatial pyramid pooling module and stacked hourglass 3D CNNs to incorporate global context information. GwcNet [13] constructed cost volume with both group-wise correlation and concatenation. Differentiable semi-global aggregation [47, 48], attention mechanism [41, 43], transformers [23] and iterative disparity range pruning [11, 37, 34, 35, 16] were also adopted in stereo matching pipeline for better generalization ability, efficiency and accuracy. RAFT-Stereo [24], CREStereo [21], IGEV [42] and DLNR [52] estimated disparity by iterative refinement using recurrent neural networks.

### 2.2 Domain Generalized Stereo Matching

In recent years, there is increasing attention towards developing stereo matching networks with domain generalization capabilities, particularly focusing synthetic-to-real generalization [48, 34, 24, 42, 16]. DSMNet [48] proposed the domain normalization layer and the trainable non-local graph-based filter to capture robust structural and geometric representations. CFNet [34] introduced a cascaded and fused multi-scale cost volume for robust stereo matching. Jing *et al.*proposed an uncertainty-guided adaptive correlation module to robustly adapt stereo matching for different scenarios [16].

Another line of research aims to improve synthetic-to-real generalization for existing stereo matching networks directly. MS-Net [2] suggested using traditional feature descriptors to build domain-invariant matching space for stereo matching. GraftNet [25] leveraged robust broad-spectrum features pre-trained on ImageNet and a feature adaptor to improve the generalization ability. FC-Stereo [49] used a stereo contrastive loss and stereo selective whitening loss to encourage stereo feature consistency across different domains. ITSA [7] utilized an information-theoretic approach to avoid short-cut learning in stereo matching. HVT [4] proposed hierarchical visual transformations to learn shortcut-invariant robust representation from synthetic images. *Our work falls within this category.*

### 2.3 Contrastive Learning

Contrastive learning trains visual representations by pulling features of positive pairs (*i.e.*, features represent the same instance, same class, or with the same defined attribute) closer and pushing negative pairs further apart. Unsupervised contrastive learning paradigms like SimCLR [5] and MOCO [14, 6] have demonstrated the potential of learned features in generalizing to various downstream tasks. Recent works in computer vision have extended contrastive learning paradigm to dense vision tasks [39, 40] in a supervised [20, 38], semi-supervised [1], or weakly-supervised manner [10, 8].

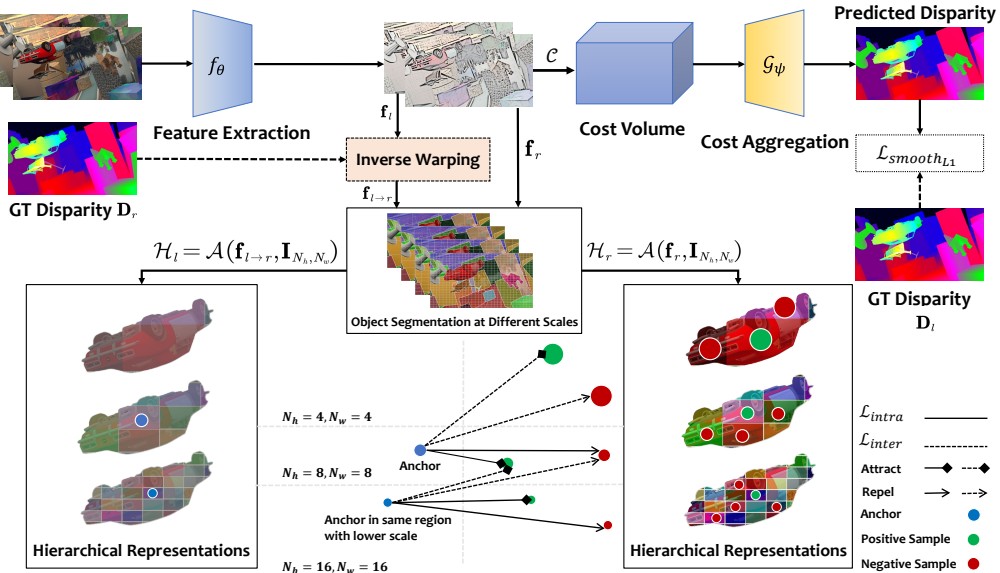

Figure 2: Overall illustration of the proposed **hierarchical object-aware dual-level contrastive learning (HODC)** framework. An example of hierarchical segmentation of a car within the input image is demonstrated, alongside with the illustration of inter- and intra-scale positive and negative pairs for dual-level contrastive learning.

Inspired by Zhang *et al.* [49] that used stereo contrastive loss to enhance feature consistency between left-right views, we further propose hierarchical dual-level contrastive learning to establish semantic and structural correspondence for generalizable stereo matching.

## 3 Method

### 3.1 Preliminaries

Given a rectified RGB stereo image pair $\mathbf{X}_{l,r} \in \mathbb{R}^{3 \times H \times W}$, stereo matching predicts horizontal displacement $\mathbf{D} \in \mathbb{R}^{H \times W}$ for every pixel in the left image $\mathbf{X}_l$. A typical stereo matching pipeline $\mathcal{F}_{\Theta}(\cdot, \cdot)$ can be written as:

$$\widehat{\mathbf{D}}_l = \mathcal{F}_{\Theta}(\mathbf{X}_l, \mathbf{X}_r) = \mathcal{G}_{\psi}\bigg( \mathcal{C}\Big( f_{\theta}(\mathbf{X}_l), f_{\theta}(\mathbf{X}_r) \Big) \bigg), \tag{1}$$

where $f_{\theta}$ denotes parameterized feature extraction network, $\mathcal{C}$ denotes cost volume construction typically via concatenation [19, 3], correlation [27, 24] or both [13, 34], and $\mathcal{G}_{\psi}$ denotes parameterized cost aggregation network and disparity regression with the soft-argmin operation [19].

Our work aims to tackle the synthetic-to-real generalization problem for stereo matching networks, where only a *synthetic* training dataset $\mathcal{D}$ is available. Here, $\mathcal{D}$ consists of stereo image pairs $\{\mathbf{X}_{l,r}^{(i)} \in \mathbb{R}^{3 \times H \times W}\}_{i=1}^{|\mathcal{D}|}$ and corresponding disparity maps $\{\mathbf{D}_{l,r}^{(i)} \in \mathbb{R}^{H \times W}\}_{i=1}^{|\mathcal{D}|}$. The realistic test data is strictly inaccessible during training. Additionally, we denote the object index map of $\mathcal{D}$ as $\{\mathbf{I}_{l,r}^{(i)} \in \mathbb{N}^{H \times W}\}_{i=1}^{|\mathcal{D}|}$, with each entry denoting the instance ID of the corresponding pixel.

### 3.2 Overall Pipeline

As illustrated in Fig. 2, our method operates on the left and right feature maps $\mathbf{f}_{l,r} = f_{\theta}(\mathbf{X}_{l,r})$ extracted by the feature extraction convolutional network $f_{\theta}(\cdot)$ within the stereo matching pipeline. Our objective is to enhance $f_{\theta}(\cdot)$ in order to extract more effective feature maps that allow semantically and structurally driven matching for domain generalizable stereo matching.

Existing stereo matching methods primarily focus on establishing pixel-wise correspondence from the left image to the right, which is prone to learning short-cut features based on appearance cues [7, 4], as they do not exhibit intrinsic knowledge such as semantic structure directly. Intuitively, enhancing semantic and structural awareness can be achieved by finding matching between corresponding object-aware regions (*i.e.*, object parts) in left-right images. These regions could vary in scale but should embody certain semantic meanings (*i.e.*, much larger than a pixel), and possess a moderate size to guide stereo matching as the ambiguities of stereo matching typically occur within a certain local area. This approach promotes semantic awareness in models, allowing regions to interact directly at different scales to explore structural information. It is supported by prior works which demonstrate that incorporating higher-level knowledge [44, 36] and promoting global awareness [3, 34] are crucial for improving the performance of stereo matching networks.

Based on this intuition, it is imperative to generate object-aware regions on the feature map with flexibility in scale, and enhance the matching between them to effectively establish semantic and structural correspondence. The pipeline of HODC consists of the following steps:

(1) In Sec. 3.3, we propose a segmentation scheme on object index to flexibly generate object-aware regions. We segment the original object index map $\mathbf{I}$ using rectangular grids with adjustable height $H/N_h$ and width $W/N_w$ to obtain sub-object level *region index* map $\mathbf{I}_{N_h,N_w}$ at any desired scale. Leveraging the sub-object level *region index* map and the extracted feature maps, we generate object-aware regional representations for both left and right views by aggregating pixel representations.

(2) In Sec. 3.4, instead of directly finding correspondence of regional representations, we first warp the left feature map $\mathbf{f}_l$ to the right using inverse warping guided by the ground-truth disparity of the right image $\mathbf{D}_r$ to obtain $\mathbf{f}_{l\to r}$. Utilizing $\mathbf{f}_{l\to r}$, $\mathbf{f}_r$ and *region index* map $\mathbf{I}_{N_h,N_w}$, we construct positive and negative pairs to represent correspondence. Adjusting the segmentation *scale* $(N_h, N_w)$ enables us to build positive and negative pairs with multi-scale, facilitating hierarchical correspondence. Furthermore, we not only establish correspondence within representations of the same scale, but also inter-scale correspondence to enhance global feature awareness.

(3) In Sec. 3.5, we introduce intra- and inter-scale contrastive loss $\mathcal{L}_{intra}, \mathcal{L}_{inter}$ using intra- and inter-scale positive and negative pairs along with the origin smooth L1 loss $\mathcal{L}_{smooth_{L1}}$, to optimize the stereo matching network end-to-end.

### 3.3 Hierarchical Object-Aware Regional Representations

Given an arbitrary feature map $\mathbf{f} \in \mathbb{R}^{C \times H \times W}$ and its corresponding object index map $\mathbf{I} \in \mathbb{N}^{H \times W}$, we can derive object-aware regional representations by computing the mean representation of pixels with the same *region index*, where *region index* denotes the segmentation result at **sub-object level**.

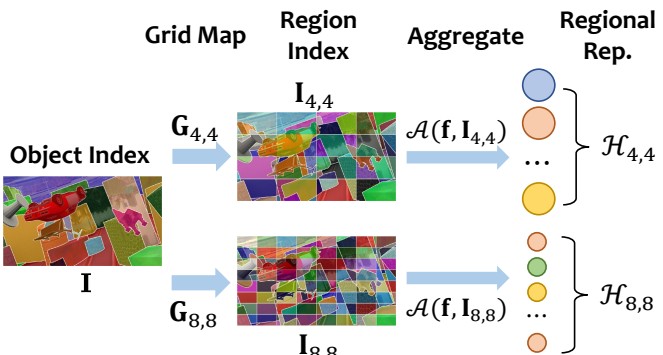

To obtain such *region index* map, we need to further perform segmentation on the given object index map. We describe the segmentation scheme by its *scale*, which can be defined by a tuple $(N_h, N_w)$. The sub-object level region index $\mathbf{I}_{N_h,N_w} \in \mathbb{N}^{H \times W}$ under scale $(N_h, N_w)$ can be derived by the following steps:

Figure 3: Illustration of generating regional representations for HODC with the feature map $\mathbf{f}$ and object index map $\mathbf{I}$ under scale $(4, 4)$ and $(8, 8)$. Different color on the image denotes pixels with different indexes, and regional representations are generated by aggregating the features of pixels with the same *region index*.

**1)** We partition the feature map into $N_h \times N_w$ rectangular grids of size $\frac{H}{N_h} \times \frac{W}{N_w}$ by dividing it horizontally and vertically, numbering these grids from 1 to $N_h \times N_w$. **2)** We then index the pixels in each grid to their corresponding grid number and denote the new grid map as $\mathbf{G}_{N_h,N_w} \in \{1, 2, \dots, N_h \times N_w\}^{H \times W}$. **3)** Using $\mathbf{I}$ and $\mathbf{G}_{N_h,N_w}$, we determine the region index $\mathbf{I}_{N_h,N_w}$ at scale

$(N_h, N_w)$ according to the following rule: pixels are assigned the same region index **if and only if** they belong to both the same object and the same grid (as illustrated in Fig. 3).

With the derived region index map $\mathbf{I}_{N_h, N_w}$ and the extracted feature map $\mathbf{f}$, we employ mean aggregation $\mathcal{A}(\cdot, \cdot)$ to obtain object-aware regional representations $\mathcal{H}_{N_h, N_w}$:

$$\mathcal{H}_{N_h, N_w} = \mathcal{A}(\mathbf{f}, \mathbf{I}_{N_h, N_w}) = \left\{ \mathcal{H}^{(i)} \;\middle|\; \mathcal{H}^{(i)} = \frac{\sum\limits_{(x,y)\,:\,\mathbf{I}_{N_h, N_w}(x,y)=i} \mathbf{f}(x, y)}{\sum\limits_{(x,y)} \mathbb{1}[\mathbf{I}_{N_h, N_w}(x, y) = i]} \right\}. \tag{2}$$

By adjusting $N_h$ and $N_w$, we can flexibly control the segmentation scale, thereby obtaining **hierarchical** object-aware regional representations with different scales.

### 3.4 Intra- and Inter-Scale Positive and Negative Pairs

**Inverse Warping and Non-Matching Region Removal.** To build positive and negative pairs, we first establish accurate correspondence for every pixel. We perform inverse warping to align the reference (left) feature map $\mathbf{f}_l$ with the target (right) $\mathbf{f}_r$ feature map using ground truth disparity of the target image:

$$\mathbf{f}_{l \to r} = \text{InvWarp}(\mathbf{f}_l, \mathbf{D}_r). \tag{3}$$

Next, we remove the non-matching pixels in the right image by left-right geometric consistency check [49], applying the reprojection error constraint and exclude them in calculating regional representations in Eq. 2. Non-matching pixels are those that appear in the target image but have no corresponding pixels in the reference image due to occlusion. The reprojection error $\mathbf{R}$ for the right image is defined as the pixel difference obtained when performing inverse warping from right to left, and then from left to right. If the reprojection error exceeds a threshold $\delta$, the pixel is considered to be occluded [49]. Note that by performing inverse warping and removing non-matching regions, we expect to find correspondence of **all pixels** that appear in both left and right views, which also indicates that the direction of warping will not influence the result.

**Intra-Scale Pairs.** With the given scale $(N_h, N_w)$, along with the warped reference feature $\mathbf{f}_{l \to r}$ and target feature $\mathbf{f}_r$, we obtain object-aware regional representation for both reference image $\mathcal{H}^l_{N_h, N_w} = \mathcal{A}(\mathbf{f}_{l \to r}, \mathbf{I}_{N_h, N_w})$ and target image $\mathcal{H}^r_{N_h, N_w} = \mathcal{A}(\mathbf{f}_r, \mathbf{I}_{N_h, N_w})$ using Eq. 2. By choosing $\mathcal{Q} = \mathcal{H}^l_{N_h, N_w}$ as the anchor (query) set and $\mathcal{K} = \mathcal{H}^r_{N_h, N_w}$ as the key set, the *intra-scale* positive sample $\mathcal{P}_{intra}$ and negative samples $\mathcal{N}_{intra}$ for the $i$-th element in $\mathcal{Q}$ can be defined as:

$$\mathcal{P}_{intra}(\mathcal{Q}^{(i)}) = \mathcal{K}^{(i)}, \mathcal{N}_{intra}(\mathcal{Q}^{(i)}) = \{\mathcal{K}^{(j)} | j \neq i\}. \tag{4}$$

**Inter-Scale Pairs.** The *inter-scale* pairs are further introduced to enhance global awareness of local representations by pulling local representation towards the corresponding region with a larger scale and away from other regions. To ensure stability during the learning process and prevent collapse, we operate in a hierarchical manner where local representations only interact with regions whose scale is within a factor $K = 4$. Regarding $(N_h, N_w)$ as the global scale, the local scale can be written as $((N_h \times k), (N_w \times k))$, where $k \leq K$ is the inter-scale factor. For instance, when the global scale is set to $(N_h, N_w) = (2, 4)$ with an inter-scale factor $k = 2$, the corresponding local scale is $(4, 8)$.

Similarly, we construct the anchor (query) set by aggregating local representations: $\mathcal{Q} = \mathcal{H}^l_{(N_h \times k), (N_w \times k)} = \mathcal{A}(\mathbf{f}_{l \to r}, \mathbf{I}_{(N_h \times k), (N_w \times k)})$. The *inter-scale* positive sample and negative samples of scale $((N_h \times k), (N_w \times k))$ are the corresponding and non-corresponding regions at the global scale of $(N_h, N_w)$, respectively, which can be obtained by the following rule:

$$\mathcal{K} = \widehat{\mathcal{H}}^r_{(N_h \times k), (N_w \times k)} = \mathcal{A}(\widehat{\mathbf{f}}_r, \mathbf{I}_{(N_h \times k), (N_w \times k)}), \tag{5}$$

$$\mathcal{P}^k_{inter}(\mathcal{Q}^{(i)}) = \mathcal{K}^{(i)}, \mathcal{N}^k_{inter}(\mathcal{Q}^{(i)}) = \{\mathcal{K}^{(j)} | j \neq i\}, \tag{6}$$

where $\widehat{\mathbf{f}}_r \in \mathbb{R}^{C \times H \times W}$ denotes the regional representation (under the global scale $(N_h, N_w)$) that each pixel lies in:

$$\widehat{\mathbf{f}}_r = \mathcal{A}^{-1}(\mathcal{H}^r_{N_h, N_w}, \mathbf{I}_{N_h, N_w}). \tag{7}$$

In Eq. 7, $\mathcal{A}^{-1}(\mathcal{H}, \mathbf{I}_{N_h, N_w})$ denotes inverse mapping from the aggregated regional representations $\mathcal{H}$ to the feature map where each pixel's value is the representation of its corresponding region.

**Hard Negative Selection.** Previous studies [18, 20, 38] have demonstrated that hard negatives that are similar to the anchor, can contribute more to the discriminating power of the learned representations, thereby reducing local ambiguities in stereo matching. Inspired by this, we select the top $10\%$ most similar negatives in $\mathcal{N}_{inter}$ and $\mathcal{N}_{intra}$ to calculate the contrastive loss.

**Remark.** It's important to note that we do not use memory banks or asymmetric structural design [14, 49] to obtain positive or negative pairs. Instead, all positive and negative pairs are calculated within the same batch. This is due to the effectiveness of using hierarchical regional representations for finding semantically and structurally driven matching.

### 3.5 Hierarchical Dual-Level Contrast and Learning Objectives

We perform hierarchical contrastive learning by employing queries and keys at different scales. Given scale $(N_h, N_w)$ and the inter-scale factor $k$, the intra-scale contrastive loss $\mathcal{L}_{intra}$ and inter-scale contrastive loss $\mathcal{L}_{inter}$ can be written as:

$$\mathcal{L}_{intra}^{(N_h, N_w), k} = \mathcal{L}\big(\mathcal{H}_{N_h, N_w}^l, \mathcal{P}_{intra}, \mathcal{N}_{intra}\big) + \mathcal{L}\big(\mathcal{H}_{(N_h \times k),(N_w \times k)}^l, \mathcal{P}_{intra}, \mathcal{N}_{intra}\big), \quad (8)$$

$$\mathcal{L}_{inter}^{(N_h, N_w), k} = \mathcal{L}\big(\mathcal{H}_{(N_h \times k),(N_w \times k)}^l, \mathcal{P}_{inter}^k, \mathcal{N}_{inter}^k\big). \quad (9)$$

To enable building more flexible representations, we do not enforce $N_h = N_w$, but instead impose weaker constraint (*i.e.*, $N_h, N_w \in \{2^i | i \in \mathbb{N}\}$) to stabilize the training process. Additionally, we set the maximum scale as $M$ to reduce the computational cost, ensuring that $N_h \times N_w \leq M$. Considering inter-scale factor $k \leq K$ as mentioned in Sec. 3.4, the dual-level hierarchical contrastive learning objectives can be formulated as:

$$\mathcal{L}_{contrastive} = \sum_{N_h, N_w \in \{2^i | i \in \mathbb{N}\}}^{N_h \times N_w \leq M} \sum_{k \in \mathbb{N}^*}^{k \leq K} \Big(\mathcal{L}_{intra}^{(N_h, N_w), k} + \mathcal{L}_{inter}^{(N_h, N_w), k}\Big). \quad (10)$$

We utilize the widely adopted InfoNCE [29] loss to conduct intra- and inter-scale contrastive learning:

$$\mathcal{L}(\mathcal{H}, \mathcal{P}, \mathcal{N}) = \frac{1}{|\mathcal{H}|} \sum_{z \in \mathcal{H}} -\log \frac{\exp(z \cdot \mathcal{P}(z)/\tau)}{\exp(z \cdot \mathcal{P}(z)/\tau) + \sum_{z_- \in \mathcal{N}(z)} \exp(z \cdot z_-/\tau)}, \quad (11)$$

where $\tau = 0.05$ denotes the temperature parameter. Together with the widely adopted smooth-L1 loss [3] $\mathcal{L}_{smooth_{L1}}$, the overall learning objective is:

$$\mathcal{L}_{overall} = \mathcal{L}_{smooth_{L1}} + \lambda \cdot \mathcal{L}_{contrastive}, \quad (12)$$

where $\lambda$ balances the weight between the smooth-L1 loss and contrastive loss.

## 4 Experiments

### 4.1 Experiment Settings

We select four stereo matching networks with different architectures as baselines to validate the performance of HODC, including two widely studied models (PSMNet [3] and GwcNet [13]), a robust method with cascaded and fused cost volume (CFNet [34]) and a recently proposed iterative-based state-of-the-art method (IGEV [42]). We integrate HODC directly during their training stage and test their generalization performance on realistic datasets.

We train all models with *synthetic* dataset SceneFlow [27] and evaluate their generalization performance on the training set of four *realistic* datasets: KITTI-2012 [12], KITTI-2015 [28], Middlebury [32] and ETH3D [33]. We use the object index map provided by the SceneFlow dataset, which can also be collected by using a pre-trained segmentation model (more discussion in Sec. 4.4). Following previous works [7, 49], we evaluate the performance of our model using the D1 metric, which calculates the percentage of outliers in the reference image with different pixel threshold $\rho$. The threshold is set to 3px for KITTI-2012 and KITTI-2015, 2px for Middlebury, and 1px for ETH3D, as suggested by dataset providers and previous works.

Table 1: Overall generalization performance comparison with previous works.

| Baseline | Method | KITTI | | Middlebury | | ETH3D | Venue |
|---|---|---|---|---|---|---|---|
| | | 2015 | 2012 | Half | Quarter | | |
| | STTR [23] | 6.7 | 8.7 | 15.5 | 9.7 | 17.2 | ICCV'2021 |
| | DSMNet [48] | 6.5 | 6.2 | 13.8 | 8.1 | 6.2 | ECCV'2020 |
| | FC-GANet [49] | 5.3 | 4.6 | 10.2 | 7.8 | 5.8 | CVPR'2022 |
| | PCWNet [35] | 5.6 | 4.2 | 15.8 | - | 5.2 | ECCV'2022 |
| | CREStereo++ [16] | 5.2 | 4.7 | - | - | 4.4 | ICCV'2023 |
| PSMNet | PSMNet [3] | 16.3 | 15.1 | 26.9 | 20.0 | 23.8 | CVPR'2018 |
| | MS-PSMNet [2] | 13.9 | 7.8 | 19.9 | 10.8 | 16.8 | 3DV'2020 |
| | FC-PSMNet [49] | 5.8 | 5.3 | 15.1 | 9.3 | 9.5 | CVPR'2022 |
| | ITSA-PSMNet [7] | 5.8 | 5.2 | 12.7 | 9.6 | 9.8 | CVPR'2022 |
| | Graft-PSMNet [25] | 4.8 | 4.3 | 9.7 | - | 7.7 | CVPR'2022 |
| | HVT-PSMNet [4] | 4.9 | 4.3 | 10.2 | - | 6.9 | CVPR'2023 |
| | **HODC-PSMNet** | **4.7** | **3.9** | **9.3** | **7.0** | **5.4** | **Ours** |
| GwcNet | GwcNet [13] | 22.7 | 20.2 | 34.2 | 18.1 | 30.1 | CVPR'2019 |
| | FC-GwcNet [49] | 5.4 | 5.0 | 11.2 | 8.2 | 8.0 | CVPR'2022 |
| | ITSA-GwcNet [7] | 5.4 | 4.9 | 11.4 | 9.3 | 7.1 | CVPR'2022 |
| | HVT-GwcNet [4] | 5.0 | **3.9** | 10.3 | - | 5.9 | CVPR'2023 |
| | **HODC-GwcNet** | **4.9** | **3.9** | **8.4** | **5.8** | **4.3** | **Ours** |
| CFNet | CFNet [34] | 5.8 | 4.7 | 15.3 | 9.8 | 5.8 | CVPR'2021 |
| | ITSA-CFNet [7] | **4.7** | 4.2 | 10.4 | 8.5 | 5.1 | CVPR'2022 |
| | HVT-CFNet [4] | 4.9 | 4.0 | 10.2 | - | 4.5 | CVPR'2023 |
| | **HODC-CFNet** | 4.8 | **3.8** | **9.5** | **7.5** | **4.2** | **Ours** |
| IGEV | IGEV [42] | 5.6 | 5.1 | 7.1 | 6.2 | 3.6 | CVPR'2023 |
| | **HODC-IGEV** | **4.5** | **3.8** | **7.0** | **5.2** | **2.7** | **Ours** |

For comparison, we include recently proposed domain generalization approaches for stereo matching networks (MS-Net [2], FCStereo [49], ITSA [7], GraftNet [25] and HVT [4]) as well as other robust architectures [23, 48, 35, 16].

## 4.2 Overall Generalization Performance Comparison

As shown in Tab. 1, after integrating HODC, the synthetic-to-real generalization performance of all baselines substantially increases by a visible margin. Compared to other generalization methods, our approach achieves the highest generalization performance in almost every setting (except for ITSA-CFNet on KITTI-2015), which demonstrates the effectiveness of HODC. HODC is also compatible with CFNet [34] using specially designed robust architecture and IGEV [42] with iterative refinement.

Notably, the generalization performance of older baselines (PSMNet, GwcNet) is comparable with or even outperforms current SOTA methods training with HODC. This demonstrates the importance of extracting semantically and structurally aware features to improve generalization for stereo matching.

## 4.3 Semantically and Structurally Driven Matching

In this section, we validate the semantically and structurally driven attributes of HODC through visualizations on the realistic Middlebury [32] and ETH3D [33] datasets. As discussed in Sec. 3.2, we select a small area in the reference image and calculate its representation to perform the query, ensuring that HODC is capable of performing accurate matching to guide the stereo matching network. We measure the similarity between this representation and the pixel representations in the target image. As shown in Fig. 4, PSMNet [3] exhibits ambiguity in matching by superficial chromatic features, resulting in high feature similarity in continuous local areas. FC-PSMNet [49] also lacks the ability to identify semantical and structural elements, showing high feature similarity in non-matching regions. Conversely, our approach accurately identifies matched regions, with limited ambiguities mainly observed in vertical areas that are not the matching candidates for stereo matching.

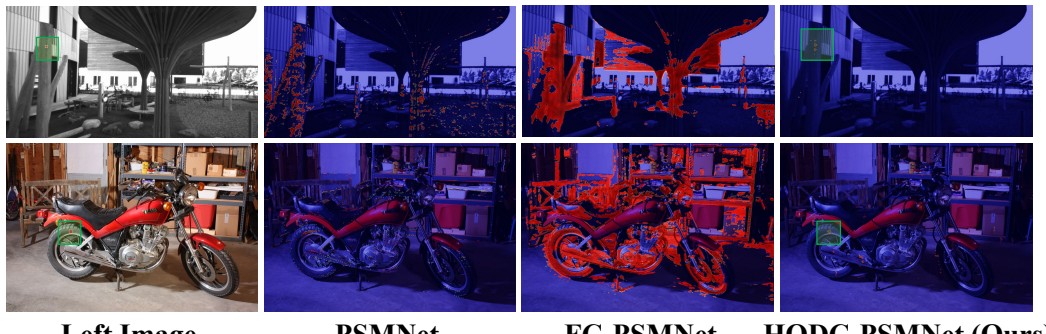

| **Left Image** | **PSMNet** | **FC-PSMNet** | **HODC-PSMNet (Ours)** |

Figure 4: Qualitative result of semantically and structurally driven matching on realistic Middlebury [32] and ETH3D [33] datasets. The first column shows the query region, which is marked red and highlighted with green box. The remaining columns show the region on the target image that has cosine similarity larger than the threshold $\alpha = 0.9$ with different methods.

Table 2: Cosine similarity comparison of positive and negative pairs on SceneFlow [27] test set.

| **Method** | **Scale** | **Intra-Scale** | | | | **Inter-Scale** | | | |
|---|---|---|---|---|---|---|---|---|---|
| | | *Pos* ↑ | *Neg* ↓ | *10% Hard* ↓ | *5% Hard* ↓ | *Pos* ↑ | *Neg* ↓ | *10% Hard* ↓ | *5% Hard* ↓ |
| PSMNet [3] | $4 \times 4$ | 0.96 | 0.47 | 0.85 | 0.88 | 0.86 | 0.47 | 0.84 | 0.87 |
| | $8 \times 8$ | 0.95 | 0.45 | 0.84 | 0.87 | 0.85 | 0.45 | 0.83 | 0.86 |
| | $16 \times 16$ | 0.95 | 0.43 | 0.83 | 0.87 | 0.83 | 0.41 | 0.80 | 0.84 |
| FC-PSMNet [49] | $4 \times 4$ | 0.99 | 0.88 | 0.97 | 0.98 | 0.97 | 0.88 | 0.97 | 0.98 |
| | $8 \times 8$ | 0.99 | 0.87 | 0.97 | 0.98 | 0.97 | 0.87 | 0.97 | 0.97 |
| | $16 \times 16$ | 0.99 | 0.86 | 0.97 | 0.97 | 0.97 | 0.85 | 0.96 | 0.97 |
| **HODC-PSMNet (Ours)** | $4 \times 4$ | 0.95 | 0.27 | 0.66 | 0.71 | 0.78 | 0.26 | 0.64 | 0.70 |
| | $8 \times 8$ | 0.95 | 0.24 | 0.62 | 0.68 | 0.78 | 0.24 | 0.61 | 0.67 |
| | $16 \times 16$ | 0.94 | 0.22 | 0.60 | 0.65 | 0.78 | 0.21 | 0.58 | 0.62 |

We further provide deeper insights into feature similarity using SceneFlow [27] test set (randomly select 500 image pairs) with dense annotations. We compare the feature cosine similarity of PSM-Net [3], FC-PSMNet [49] and HODC-PSMNet between matching intra- and inter-scale regions at different scales. Additionally, we include the cosine similarity of the top 5% and 10% hardest negative paris. Zhang *et al.* [49] suggest that feature consistency between matching pixels is consistent with the model's generalization performance. However, referring to Tab. 2, we find that dissimilarity between negatives that can provide more discriminative power plays a more important role in the model's generalization ability, which is neglected in previous works. Using the HODC framework, the hardest negatives at both intra- and inter-scale can also be effectively distinguished.

## 4.4 Segmentation Scale Analysis and Ablation Study

In this section, we conduct further analysis of HODC on PSMNet [3] and GwcNet [13]. First, we investigate the effect of the segmentation scale by changing $M$ in Eq. 10 (keeping $K = 4$ unchanged). As shown in Tab. 3, the performance of our method remains stable with different $M$ in a relatively large interval. Notably, segmenting the object using fewer grids to perform contrastive learning on a larger scale can improve the performance of stereo matching networks on certain datasets (*e.g.*, KITTI-2012 and KITTI-2015), while other datasets or networks benefit from a larger $M$ to enhance performance. This can be attributed to larger $M$ providing more accurate local matching with more negative samples and finer representations, while smaller $M$ derives features on larger scales.

Next, we perform an ablation study to examine the effectiveness of intra- and inter-scale contrastive loss. Referring to Tab. 3, the performance of PSMNet and GwcNet decreases on all datasets without using $\mathcal{L}_{intra}$ or $\mathcal{L}_{inter}$. Furthermore, we analyze the robustness of the proposed dual-level contrastive learning with hierarchical representations when the object index map is inaccurate. We conduct an extreme case evaluation by omitting the object prior and performing segmentation directly on the entire image. Though a decline in performance is observed, the proposed method still achieves

Table 3: Analysis of segmentation scale and ablation study of the proposed framework.

| Network | $M$ | $\mathcal{L}_{intra}$ | $\mathcal{L}_{inter}$ | Object | KT-15 | KT-12 | Mid-H | Mid-Q | ETH3D |
|---|---|---|---|---|---|---|---|---|---|
| PSMNet | 32 | ✓ | ✓ | ✓ | **4.6** | 4.0 | 9.8 | 7.4 | **4.9** |
|  | 64 | ✓ | ✓ | ✓ | **4.6** | **3.8** | 9.5 | 7.4 | 5.8 |
|  | 128 | ✓ | ✓ | ✓ | 4.7 | 3.9 | **9.3** | **7.0** | 5.4 |
|  | 128 | × | ✓ | ✓ | 4.9 | 4.0 | 9.7 | 7.2 | 5.7 |
|  | 128 | ✓ | × | ✓ | 4.9 | 4.1 | 9.9 | 7.9 | 5.9 |
|  | 128 | ✓ | ✓ | × | 4.9 | 4.2 | 10.6 | 8.1 | 6.6 |
| GwcNet | 32 | ✓ | ✓ | ✓ | **4.8** | **3.8** | 8.9 | **5.4** | 5.0 |
|  | 64 | ✓ | ✓ | ✓ | **4.8** | **3.8** | 8.6 | 5.7 | 4.4 |
|  | 128 | ✓ | ✓ | ✓ | 4.9 | 3.9 | **8.4** | 5.8 | **4.3** |
|  | 128 | × | ✓ | ✓ | 5.0 | 4.1 | 9.0 | 6.1 | 5.4 |
|  | 128 | ✓ | × | ✓ | 5.1 | 4.2 | 9.2 | 6.2 | 5.2 |
|  | 128 | ✓ | ✓ | × | 5.3 | 4.1 | 10.3 | 6.7 | 5.7 |

considerable performance and outperforms some SOTA methods (*e.g.*, FCStereo [49], ITSA [7]) by establishing local and global correspondence with our dual-level contrastive loss.

## 5   Conclusion

We proposed a novel and effective hierarchical object-aware dual-level contrastive learning (HODC) framework aimed at enhancing stereo matching networks' ability to extract semantically and structurally meaningful features and improve their generalization performance. Our approach leveraged a flexible segmentation scheme on objects to obtain hierarchical object-aware representations, followed by intra- and inter-scale contrastive learning guided by ground-truth disparity. Through extensive experiments with visualization, we have demonstrated that our method facilitates networks in achieving better generalization by enabling them to find semantically and structurally driven matches.

## Acknowledgements

This work was supported in part by NTU-DESAY SV Research Program under Grant 2018-0980.

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

# Appendix

## A  Dataset

**SceneFlow** [27] is a large scale *synthetic* dataset consisting of three subsets: FlyingThings3D, Driving, and Monkaa. In all, SceneFlow provides 35,454 training stereo image pairs and 4,370 testing image pairs with a resolution of $960 \times 540$, with dense ground-truth disparity and object index. *All our models are trained on SceneFlow training set only*.

We evaluate the generalization performance of our models on the training set of four *realistic datasets*: KITTI-2012 [12], KITTI-2015 [28], Middlebury [32] and ETH3D [33]. **KITTI-2012** [12] and **KITTI-2015** [28] contain 194 and 200 stereo pairs with sparse ground-truth disparity, respectively, collected from driving cars using LiDAR. **Middlebury** [32] includes 15 high-resolution stereo pairs with indoor scenes, and we use its half- and quarter-resolution versions in our experiments. **ETH3D** [33] provides 27 grayscale image pairs collected from both indoor and outdoor scenes.

## B  Implementation

### B.1  Implementation Details

Following [4, 49], we use instance and domain normalization [48] in the feature extraction network. For HODC-PSMNet, we use the same architecture as FC-PSMNet [49]. For HODC-GwcNet and HODC-CFNet, we replace all normalization layers with domain normalization as suggested in [4]. For HODC-IGEV, we follow their original implementation [42].

All models are implemented by Pytorch and trained with the Adam optimizer ($\beta_1 = 0.9, \beta_2 = 0.999$). We train the models from scratch (except for IGEV which we follow their original implementation using MobileNetV2 pretrained on ImageNet [9]) with a batch size of $8$ for $45$ epochs on Scene-Flow [27]. The learning rate is set to $0.001$, which decreases by half after epoch 15 and 30 (except for IGEV which we follow their original implementation [42]). The input image are normalized with the mean ($[0.485, 0.456, 0.406]$) and variation ($[0.229, 0.224, 0.225]$) of ImageNet [9]. The maximum disparity $D$ for training and evaluation is set to $D = 192$ for PSMNet, GwcNet, IGEV, and $D = 256$ for CFNet. To fairly compare with previous methods [34, 4, 49], we employ asymmetric augmentation [34] during data processing.

For all experiments, we set the reprojection error threshold to $\delta = 3$ in Sec. 3.4 for occlusion removal following [49], which is not a sensitive hyper-parameter as we use accurate ground-truth disparity map to calculate reprojection error. The maximum segmentation scale mentioned in Sec. 3.5 is set to $M = 128$, which is further discussed in Sec. 4.4. In practice, we do not calculate all possible $N_h$, $N_w$ and $k$ in Eq. 10 within a single iteration, but randomly pick $N_h$, $N_w$ and $k$ that satisfy the constraint in every iteration for efficiency. The contrastive loss weight $\lambda$ decreases from $5.0$ to $2.5$ gradually from the first epoch to the last epoch for all models. Following recent SOTA methods (*e.g.*, ITSA [7] and HVT [4]), we evaluate all pixels in the KITTI (and DrivingStereo [45]) datasets and non-occluded pixels in Middlebury and ETH3D datasets where occlusion masks are available.

### B.2  Features for HODC

It's worth noting that the spatial size of the extracted feature map $\widetilde{\mathbf{f}} = f_\theta(\mathbf{X})$ may differ from the original image size which may not align with the object index. In HODC, we choose the feature map $\widetilde{\mathbf{f}}$ with a spatial size $\frac{H}{4} \times \frac{W}{4}$ in the feature extraction network, and upsample it to the desired size using bilinear interpolation:

$$\mathbf{f}_{l \rightarrow r} = \text{Upsample}(\widetilde{\mathbf{f}}_{l \rightarrow r}), \mathbf{f}_r = \text{Upsample}(\widetilde{\mathbf{f}}_r). \qquad \text{(a)}$$

Additionally, we observe that feature maps used to construct correlation cost volume often possess a relatively large channel dimension (*e.g.*, $C = 320$ for GwcNet [13]), directly using them is inefficient. Therefore, we also provide an alternative approach to use **group-wise average value** of $\widetilde{\mathbf{f}}$, reducing the channel dimension of the feature map to $N_g$ for lower computational cost and memory usage, where $N_g$ denotes the number of groups. We formulate the entire procedure of mapping the raw

feature from the feature extraction network $\widetilde{\mathbf{f}}$ to the feature $\mathbf{f}$ used for HODC as:

$$\mathbf{f}_{corr}(c, x, y) = \text{Upsample}\left(\frac{1}{C/N_g} \sum_{i=c \times C/N_g}^{(c+1) \times C/N_g - 1} \widetilde{\mathbf{f}}_{corr}(i, x, y)\right), \tag{b}$$

$$\mathbf{f}_{cat} = \text{Upsample}(\widetilde{\mathbf{f}}_{cat}), \tag{c}$$

$$\mathbf{f} = \mathbf{f}_{corr} \| \mathbf{f}_{cat}, \tag{d}$$

where $\widetilde{\mathbf{f}}_{corr}$ and $\widetilde{\mathbf{f}}_{cat}$ denotes feature map to build correlation and concatenation cost volume in the stereo matching pipeline, respectively, and $\|$ denotes concatenation operation along the feature dimension. If only correlation or concatenation is used for building cost volume, then $\mathbf{f}$ is directly derived from Eq. b or Eq. c (*i.e.*, Eq. d is omitted). In our experiment, we do not use group-wise mean for HODC-PSMNet and HODC-IGEV. For HODC-GwcNet and HODC-CFNet, we set $N_g = 40$.

We further provide quantitative results on memory usage and computational cost. We use a single Nvidia RTX 3090 graphics card (with 24 GiB memory) and the batch size is set to 2 for the experiment. Referring to Tab. A, group-wise mean operation can effectively reduce memory consumption and computational cost while the model can still achieve state-of-the-art performance.

Table A: Analysis on computational resources.

| Method | $N_g$ | Memory Usage | Time |
|---|---|---|---|
| GwcNet [13] | - | 6.82 GiB | ∼1.8s/iter |
| HODC-GwcNet | 20 | 10.30 GiB | ∼2.5s/iter |
| HODC-GwcNet | 40 | 11.90 GiB | ∼2.7s/iter |
| HODC-GwcNet | 80 | 13.63 GiB | ∼3.2s/iter |

**Remark.** Operating at the spatial size of the original image (*i.e.*, $H \times W$) does not significantly increase the computational cost or memory usage, as the size of the hierarchical representations $|\mathcal{H}_{N_h \times N_w}|$ is determined and only determined by the *region index* map $\mathbf{I}_{N_h, N_w}$, which can be controlled by adjusting the segmentation scale $(N_h, N_w)$, thus enabling us to calculate the similarity between all representation pairs within a mini-batch. Moreover, experiments in Sec. 4.4 demonstrate that the performance is not sensitive to the segmentation scale within a relatively large interval, with competitive performance achieved using coarse-grained segmentation.

## C   Hyper-Parameter Analysis

In this section, we perform a more detailed hyper-parameter analysis on the weight $\lambda$ of our proposed dual-level contrastive loss and the proportion of the selected negative samples using PSMNet [3] and GwcNet [13]. Other experiment settings follow Sec. B.1.

### C.1   Analysis on Contrastive Loss Weight

Table B: Analysis on the relative weight of our proposed contrastive loss.

| Network | Rel. Weight | KITTI | | Middlebury | | ETH3D |
|---|---|---|---|---|---|---|
| | | 2015 | 2012 | Half | Quarter | |
| HODC-PSMNet | 1.0 | 4.7 | 3.9 | 9.8 | 7.2 | **5.0** |
| | 2.0 | **4.5** | **3.8** | 9.5 | 7.3 | 5.4 |
| | 3.0 | 4.6 | 4.0 | **9.3** | **7.0** | 6.0 |
| HODC-GwcNet | 1.0 | 4.9 | **3.8** | 8.5 | **5.3** | 4.8 |
| | 2.0 | **4.8** | **3.8** | 9.2 | 5.7 | **4.6** |
| | 3.0 | **4.8** | **3.8** | **8.5** | 6.0 | 5.1 |

The weight $\lambda$ controls the trade-off between finding matchings of object-aware regions and accurate pixel-level matching. As the weight of smooth-L1 loss is inconsistent across different baselines

(*e.g.*, there are 3 outputs in PSMNet, each with weight $(1.0, 0.7, 0.5)$), we use relative weight for consistency (*i.e.*, $\lambda$ divided by the total weight of smooth-L1 loss). As shown in Tab. B, the performance of both HODC-PSMNet and HODC-GwcNet on all datasets remain stable in a relatively large interval (with $\lambda$ varying from 1x to 3x). This demonstrates that the goal of semantically and structurally driven matching aligns with stereo matching, and HODC does not require extensive hyper-parameter tuning.

## C.2 Analysis on Hard Negative Selection

Table C: Analysis on the proportion of selected hard negatives.

| Network | Negative Proportion | KITTI | | Middlebury | | ETH3D |
|---|---|---|---|---|---|---|
| | | 2015 | 2012 | Half | Quarter | |
| HODC-PSMNet | 10% | **4.7** | **3.9** | **9.3** | **7.0** | 5.4 |
| | 30% | **4.7** | **3.9** | 9.5 | 7.1 | **4.7** |
| | 50% | **4.7** | 4.0 | 10.4 | 7.9 | 5.8 |
| HODC-GwcNet | 10% | **4.9** | **3.9** | **8.4** | 5.8 | **4.3** |
| | 30% | **4.9** | **3.9** | 8.5 | **5.3** | 4.6 |
| | 50% | **4.9** | **3.9** | 8.5 | 5.8 | 4.7 |

Selecting an appropriate amount of negative samples in contrastive learning can make the training procedure more effective and boost the performance [38]. We further investigate the impact of the selection proportion of hard negatives. As shown in Tab. C, selecting 10% to 30% hardest negatives can contribute to better performance, especially on Middlebury and ETH3D datasets.

## D  Robustness to Broader Scenarios

In this section, we further evaluate the generalization performance of training stereo networks from synthetic datasets using our approach on DrivingStereo [45] dataset. The dataset is collected on a driving car with various weather conditions: *sunny*, *cloudy*, *rainy*, and *foggy*, with 500 image pairs each, simulating complex real-world scenarios. We choose PSMNet [3] and GwcNet [13] as baselines, and compare with recently developed domain generalization technique for stereo matching [7, 49, 4]. Following [4], we also include the results of *officially released fine-tuned network* (FT-PSMNet and FT-GwcNet) on KITTI-2015 [28].

As shown in Tab. D, our method outperforms all baselines in most scenarios and achieves the best overall performance. Notably, our method outperforms the fine-tuned models on KITTI-2015 without

Table D: Robustness comparison of different methods on the DrivingStereo [45] dataset collected from diverse challenging realistic scenarios: Sunny, Cloudy, Rainy, and Foggy. We employ the D1 (3px) metric to evaluate generalization performance.

| Method | Sunny | Cloudy | Rainy | Foggy | Avg. |
|---|---|---|---|---|---|
| PSMNet [3] | 62.5 | 60.1 | 60.5 | 68.6 | 62.9 |
| FT-PSMNet [7] | 4.0 | **2.9** | 11.5 | 6.5 | 6.2 |
| FC-PSMNet [49] | 4.9 | 4.3 | **7.2** | 6.2 | 5.7 |
| ITSA-PSMNet [7] | 4.8 | 3.2 | 9.4 | 6.3 | 5.9 |
| HVT-PSMNet [4] | 4.2 | 3.1 | 8.7 | 5.6 | 5.4 |
| **HODC-PSMNet (Ours)** | **3.3** | 3.0 | 7.8 | **4.2** | **4.6** |
| GwcNet [13] | 18.1 | 24.7 | 28.2 | 28.3 | 24.8 |
| FT-GwcNet [7] | 3.1 | **2.5** | 12.3 | 6.0 | 6.0 |
| FC-GwcNet [49] | 4.7 | 5.1 | 9.1 | 7.8 | 6.7 |
| ITSA-GwcNet [7] | 4.4 | 3.3 | 9.8 | 5.9 | 5.9 |
| HVT-GwcNet [4] | 3.4 | 3.5 | 8.6 | 5.6 | 5.3 |
| **HODC-GWCNet (Ours)** | **3.0** | 2.9 | **7.7** | 4.7 | **4.6** |

exposure to realistic data, which further demonstrates the importance of semantically and structurally aware attributes of the extracted features. Visualization results are shown in Fig. A.

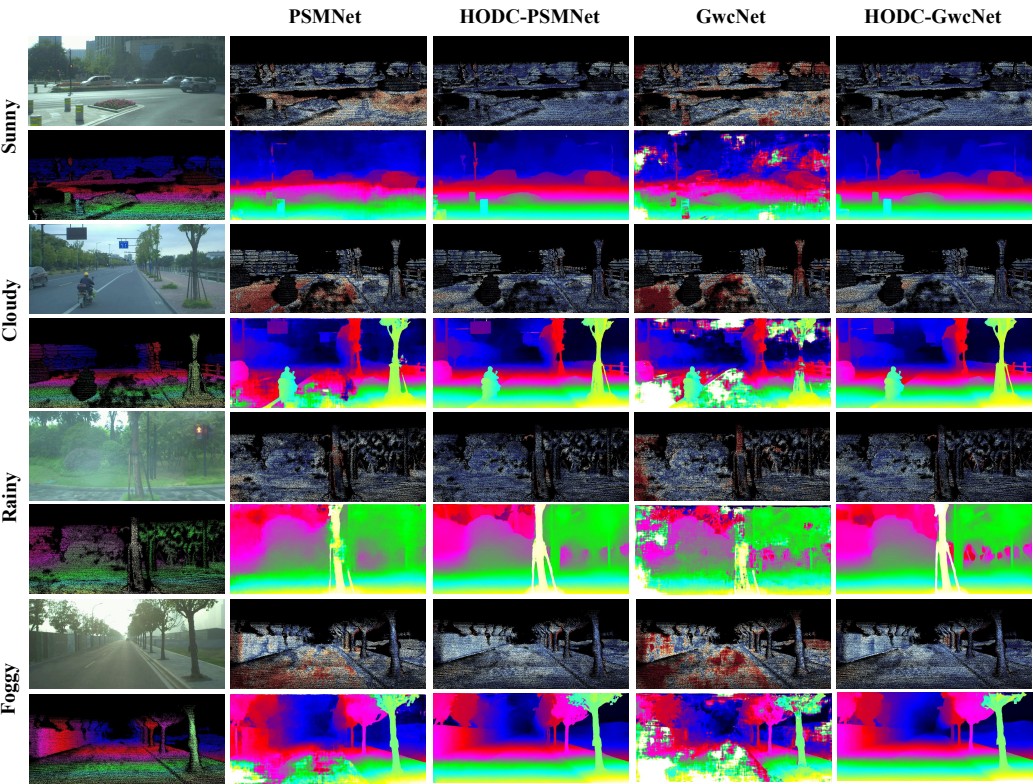

Figure A: Qualitative comparisons on the DrivingStereo [45] dataset under 4 different weather conditions. The first column shows the left image and the corresponding ground-truth disparity map. The rest columns show the error map and the predicted disparity map of PSMNet [3], HODC-PSMNet, GwcNet [13] and HODC-GwcNet, respectively.

## E    Limitations

While our method seamlessly integrates into existing stereo matching networks and is compatible with various architectures, direct integration overlooks the impact of network architectures on domain generalized stereo matching. Further enhancing generalization performance may involve designing network architecture that emphasizes semantically and structurally driven matching. As part of our future endeavors, we will also focus on designing network architectures that align with the principles established in our work.

## F    More Visualizations for Semantically and Structurally Driven Matching

In this section, we provide more visualizations to validate the semantically and structurally driven attributes of our method using PSMNet [3] on the following realistic datasets: KITTI [12, 28], Middlebury [32] and ETH3D [33]. PSMNet, FC-PSMNet [49] and HODC-PSMNet are selected for comparison. Similar to Sec. 4.3, we measure the cosine similarity of the representations between the selected region (marked red and highlighted with green box) in the reference image and every pixel in the target image. Only pixels with cosine similarity larger than the threshold $\alpha$ are shown. Referring to Fig. B, Fig. C and Fig. D, our method is capable of performing accurate semantically and structurally driven regional matching as our approach accurately identifies matched regions compared to the baselines.

| Left Image | PSMNet | FC-PSMNet | HODC-PSMNet (Ours) |

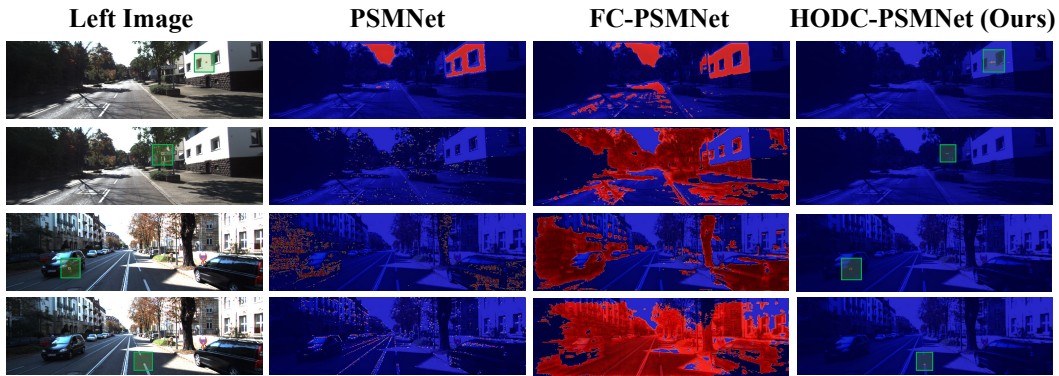

Figure B: Visualizations for semantically and structurally driven matching on the KITTI [12, 28] dataset (better view in zoomed mode.)

| Left Image | PSMNet | FC-PSMNet | HODC-PSMNet (Ours) |

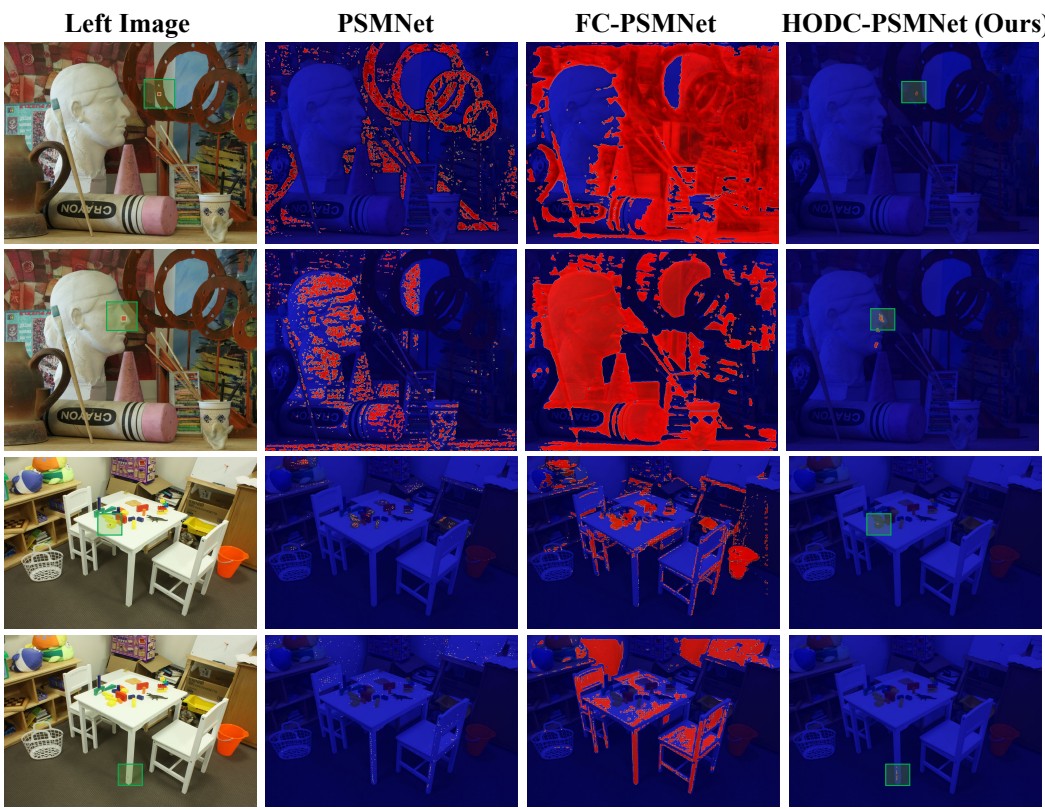

Figure C: Visualizations for semantically and structurally driven matching on the Middlebury [32] dataset (better view in zoomed mode.)

# G    More Qualitative Results

In this section, we provide more qualitative comparison between our method and the baselines (PSMNet [3], GwcNet [13], IGEV [42]) on the following realistic datasets: KITTI [12, 28], Middlebury [32] and ETH3D [33]. (Referring to Fig. E, Fig. F and Fig. G.) *Note that all models are trained only on the synthetic SceneFlow [27] training set.*

| Left Image | PSMNet | FC-PSMNet | HODC-PSMNet (Ours) |
|:---:|:---:|:---:|:---:|

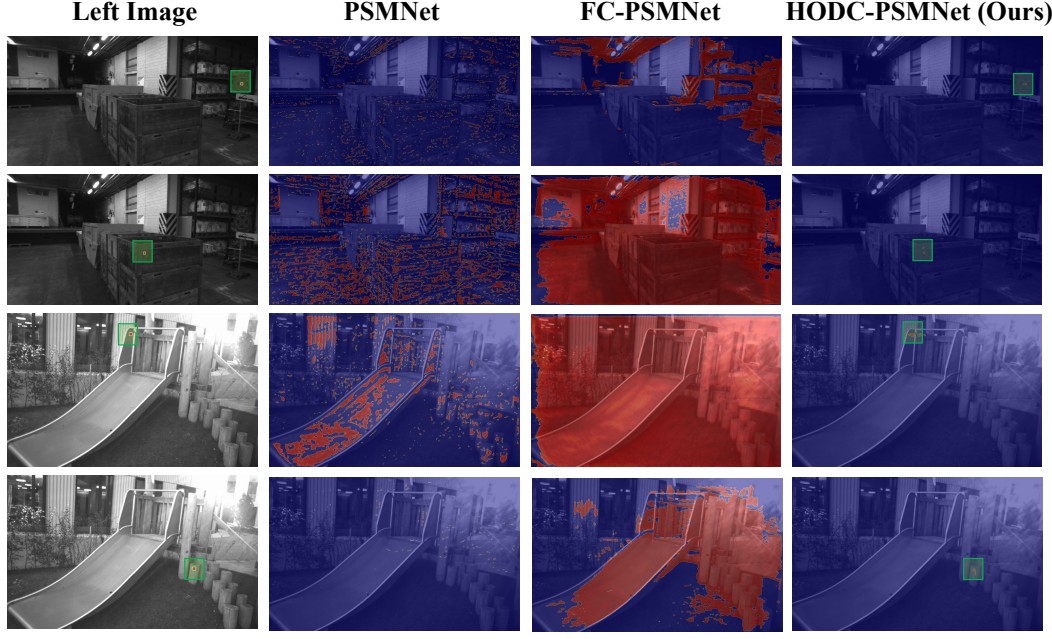

Figure D: Visualizations for semantically and structurally driven matching on the ETH3D [33] dataset (better view in zoomed mode.)

| | PSMNet | GwcNet | IGEV |
|:---:|:---:|:---:|:---:|

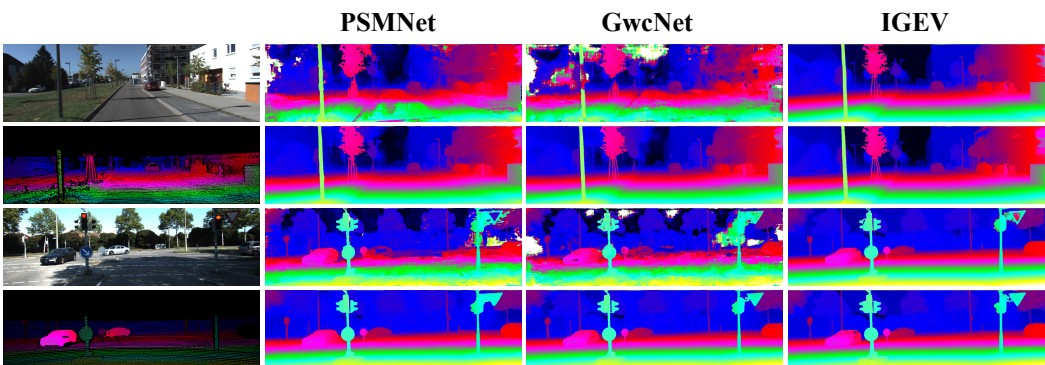

Figure E: Qualitative comparisons on the KITTI [12, 28] dataset. The first column shows the left image and the corresponding ground-truth disparity map. The rest columns compare the results of the different stereo matching networks w/ and w/o using HODC. In each scene, the first row denotes the baselines and the second row denotes the results with HODC.

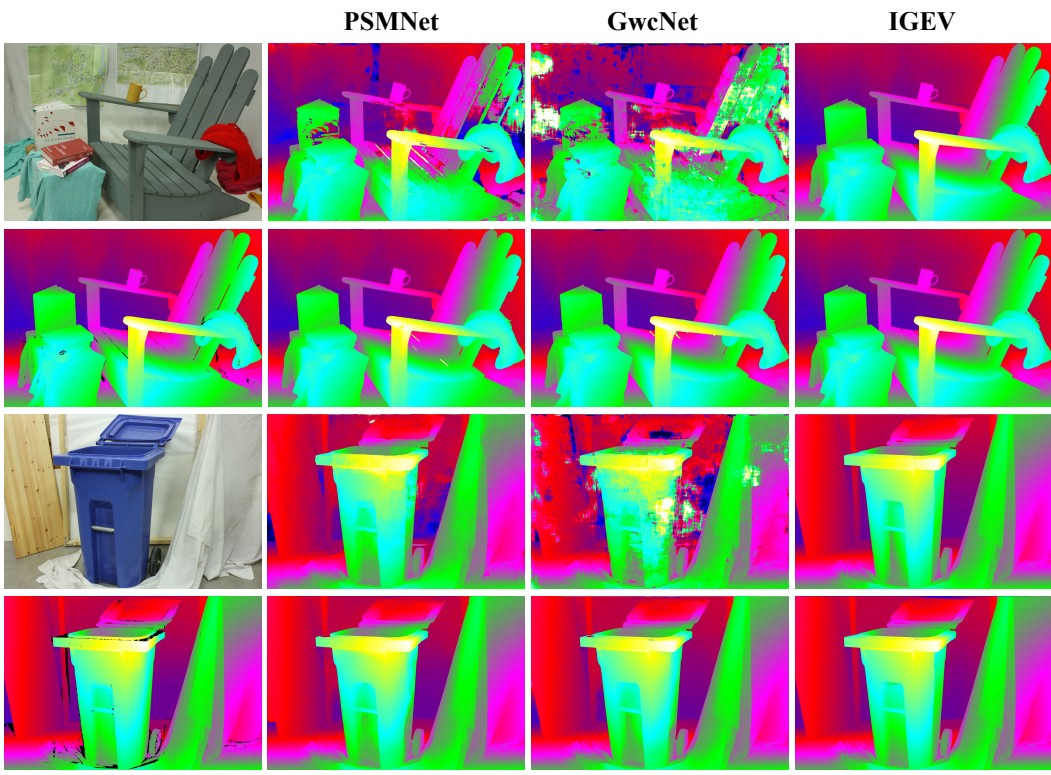

Figure F: Qualitative comparisons on the Middlebury [32] dataset. The first column shows the left image and the corresponding ground-truth disparity map. The rest columns compare the results of the different stereo matching networks w/ and w/o using HODC. In each scene, the first row denotes the baselines and the second row denotes the results with HODC.

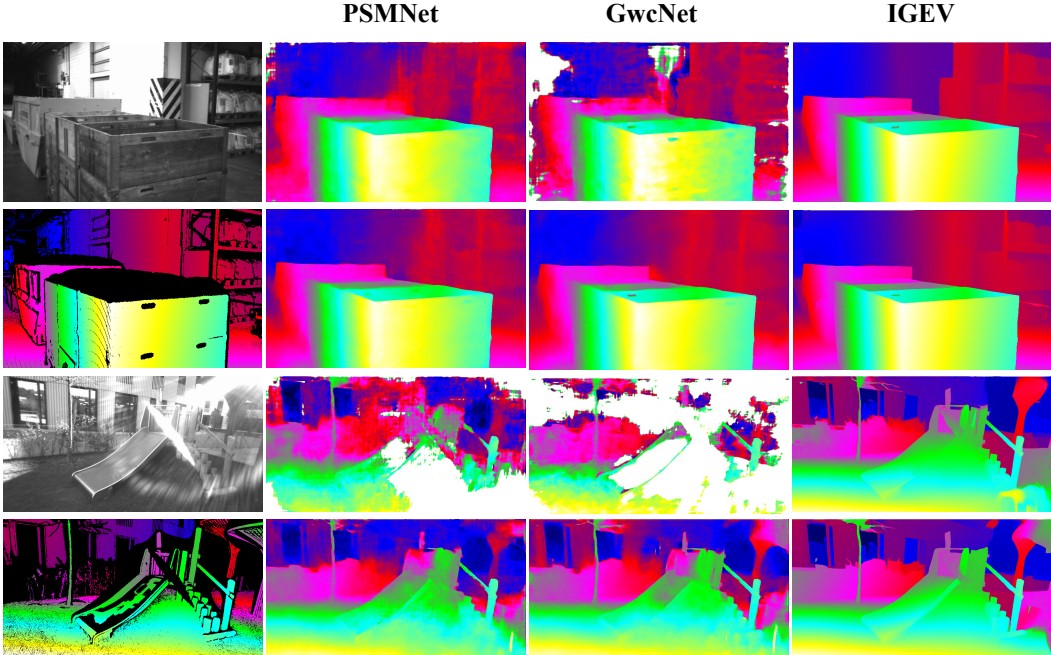

Figure G: Qualitative comparisons on the ETH3D [33] dataset. The first column shows the left image and the corresponding ground-truth disparity map. The rest columns compare the results of the different stereo matching networks w/ and w/o using HODC. In each scene, the first row denotes the baselines and the second row denotes the results with HODC.

