# OpenReview forum: "Hierarchical Object-Aware Dual-Level Contrastive Learning for Domain Generalized Stereo Matching"
_NeurIPS.cc/2024/Conference — NeurIPS 2024 poster_

### Official Review · Reviewer_ApGF · 2024-06-25

**Soundness:** 3
**Presentation:** 3
**Contribution:** 3
**Rating:** 6
**Confidence:** 4

**Summary:**

The authors propose a novel framework to achieve strong domain generalization from synthetic disparity+semantic labels datasets. To achieve this goal, the framework employs object-aware dual-level contrastive learning (HODC) to guide the backbone network toward robust and general feature extraction: these general features are the key idea to achieve good generalization results.
In particular, to achieve general feature extraction, authors present two hierarchical contrastive losses -- i.e., contrastive loss is applied at multi-scale -- that work i) at the same scale level (intra-scale); ii) at different scale levels (inter-scale).
Exhaustive experiments (qualitative and quantitative) on five real datasets confirm the superiority of the proposal w.r.t. other related domain generalization techniques.

**Strengths:**

**State-of-the-art performance**: Tab. 1 assesses the performance of the proposal against several domain generalization frameworks (i.e., MS-Net, FCStereo, ITSA, GraftNet, and HVT) and one recent SOTA stereo architecture -- i.e., IGEV. Sometimes the proposal shows an error reduction of over 1% which is remarkable. It seems that the HODC framework exhibits competitive performance also w.r.t methods that achieve domain generalization using networks trained on NERF-generated stereo datasets (R1) or guided by an external sparse depth sensor (strong assumption) (R2). Finally, Tab. D shows interesting results on another automotive dataset: the proposal confirms the superiority over other techniques and often w.r.t. the vanilla network fine-tuned in the final domain.

**Extensive experiments**: The main experiment in Tab. 1 is exhaustive -- i.e., includes other recent domain generalization techniques and a recent SOTA network. Furthermore, the authors run extensive ablation studies w.r.t. all proposed components of the framework. Notably, in Tab. 2 authors empirically demonstrate the effects of positive pairs and (more importantly) the impact of negative pairs, improving studies of the previous literature.

**The reading was smooth**: The authors wrote the paper in a linear and clear way. They expose the problem to the reader and following logical steps they arrive at the proposed solution. The figures are clear and help the reader to understand the proposal and the results. There are some minor imperfections that can be fixed (see question paragraph).

(R1) Tosi, F., et al. (2023). Nerf-supervised deep stereo. In Proceedings of the IEEE/CVF Conference on Computer Vision and Pattern Recognition (pp. 855-866).

(R2) Bartolomei, L., et al. (2023). Active stereo without pattern projector. In Proceedings of the IEEE/CVF International Conference on Computer Vision (pp. 18470-18482).

**Weaknesses:**

**The method requires semantic labels to achieve SOTA performance**: it is true that the proposal is effective even without the object prior, however, it is necessary to achieve SOTA performance: not all datasets output this information. Authors could have better studied the usage of a SOTA segmentation framework (e.g., (R3)) to replace ground-truth semantic labels.

(R3) Kirillov, A., et al. (2023). Segment anything. In Proceedings of the IEEE/CVF International Conference on Computer Vision (pp. 4015-4026).

**Questions:**

Before questions, I resume here the motivations behind my overall rating: given the text clarity, the exhaustive experiments w.r.t competitors and other studies (i.e., ablation studies and effects of negative and positive feature pairs), the requirements of additional semantic labels, my final rating is "Weak Accept".

Authors can improve the paper by showing the effects of a SOTA segmentation network to replace ground-truth labels (this could be done offline, so there are no constraints on the segmentation model). Furthermore, experiments could be extended using other recent stereo networks robust against domain shift (e.g., RAFT-Stereo (R4)).

**Minor comments**: 1) It is true that the framework does not require changes in the stereo network architecture, however, it still needs access to the model to train it with the proposed losses -- i.e., the stereo network is seen as a "white box" model. 2) Row 220: I suggest the authors add a little example of global and local scale to further reduce ambiguities (e.g., k=2, N_h = 2, N_w = 4 -> global scale is (2,4) and local scale is (4,8)). 3) Tab. 1: techniques are evaluated using all pixels or only non-occluded pixels? There is a little typo: it is ITSA, not ISTA.


(R4) Lipson, L., et al. (2021). Raft-stereo: Multilevel recurrent field transforms for stereo matching. In 2021 International Conference on 3D Vision (3DV) (pp. 218-227). IEEE.


## After Reviewer-Authors decision

After carefully reading the Authors' rebuttal, I decided to confirm my previous decision.

**Limitations:**

The authors have adequately addressed the limitations. However, I would have also added that the method requires semantic labels.

---

> ### Author Rebuttal · Authors · 2024-08-06
>
> Thanks for your positive feedback and helpful suggestions. We would like to make the following response to your questions:
>
> > Q1: The method requires semantic labels to achieve SOTA performance: it is true that the proposal is effective even without the object prior, however, it is necessary to achieve SOTA performance: not all datasets output this information. Authors could have better studied the usage of a SOTA segmentation framework (*e.g.*, (R3)) to replace ground-truth semantic labels.
>
> A1: Thanks for expressing your concerns. We would like to highlight that, in contrast to other methods that utilize additional information (*e.g.*, using pretrained features on large-scale labeled datasets [3]) to improve the generalization ability of stereo matching networks, our method relies solely on synthetic data sources. In synthetic stereo datasets, object labels can be generated along with disparity maps without much extra effort, and they are often available [4, 5]. As using pseudo labels generated by pre-trained models trained on realistic data does not strictly follow the synthetic-to-real setting, in the future we plan to directly derive pseudo object masks from disparity for finding semantically and structurally driven matching.
>
> Additionally, our HODC is orthogonal with other proposed SOTA domain generalization approaches (*e.g.*, hierarchical visual transformation [1], adversarial shortcut perturbations [2]). The proposed HODC can incorporate these methods for data augmentation, which could further improve its generalization performance.
>
> > Q2: Furthermore, experiments could be extended using other recent stereo networks robust against domain shift (e.g., RAFT-Stereo (R4)).
>
> A2: Thank you for your advice. In our experiments, we included IGEV [6], a more recent deep network architecture that serves as a strong baseline, as it incorporates RAFT-Stereo with Geometry Encoding Volume (GEV) that shows SOTA in-domain as well as generalization performance. Experiments in Table 1 show that the generalization performance of HODC-IGEV is also substantially improved compared to the baseline, demonstrating the compatibility of HODC with recent stereo networks to enhance their robustness to domain shift.
>
> > Q3: It is true that the framework does not require changes in the stereo network architecture, however, it still needs access to the model to train it with the proposed losses -- i.e., the stereo network is seen as a "white box" model.
>
> A3: Thank you for providing insights regarding the extensibility of HODC with particular constraints. We wish to highlight that our HODC operates on the intermediate stage (*i.e.*, after feature extraction) in a stereo matching pipeline, and hence it does not need to access the particular structure of the model. Furthermore, as feature extraction is a very common component in deep-learning-based stereo matching networks, it is rather straightforward to integrate HODC into existing models, thus ensuring its universality.
>
> > Q4: Row 220: I suggest the authors add a little example of global and local scale to further reduce ambiguities (e.g., k=2, N_h = 2, N_w = 4 -> global scale is (2,4) and local scale is (4,8)).
>
> A4: Thank you for the suggestion to improve our paper. We will include the suggested example in the final manuscript to reduce ambiguities.
>
> > Q5: Tab. 1: techniques are evaluated using all pixels or only non-occluded pixels?
>
> A5: Following the implementations of recent SOTA methods (e.g., ITSA and HVT), we evaluate all pixels in the KITTI (and DrivingStereo) dataset and non-occluded pixels in Middlebury and ETH3D datasets.
>
> > Q6: There is a little typo: it is ITSA, not ISTA.
>
> A6: Thank you for pointing this out. We will fix the typos in our final manuscript.
>
>
>
> **References**
>
> [1] Domain Generalized Stereo Matching via Hierarchical Visual Transformation. CVPR 2023.
>
> [2] ITSA: An Information-Theoretic Approach to Automatic Shortcut Avoidance and Domain Generalization in Stereo Matching Networks. CVPR 2022.
>
> [3] GraftNet: Towards Domain Generalized Stereo Matching with a Broad-Spectrum and Task-Oriented Feature. CVPR 2022.
>
> [4] Virtual Worlds as Proxy for Multi-Object Tracking Analysis. CVPR 2016.
>
> [5] A naturalistic open source movie for optical flow evaluation. ECCV 2012.
>
> [6] Iterative Geometry Encoding Volume for Stereo Matching. CVPR 2023.

---

> > ### Comment · Reviewer_ApGF · 2024-08-07
> >
> > Dear Authors,
> >
> > Thanks for your effort in the response. After carefully reading your rebuttal, this is my reply:
> >
> > **A1**:
> > - _in the future we plan to directly derive pseudo object masks from disparity for finding semantically and structurally driven matching._ I agree: this is a valid alternative to my suggestion.
> >
> > **A2**
> > - _...a more recent deep network architecture that serves as a strong baseline, as it incorporates RAFT-Stereo..._ I agree. However, RAFT-Stereo was just an example: there are a lot of recent SOTA stereo networks that could be inserted to enrich the paper.
> >
> > **A3**
> > - _and hence it does not need to access the particular structure of the model._ I apologize for the confusion: for "white-box model" I was referring to networks that allow i) access to feature extraction architecture; ii) back-propagation. In practice, your proposal requires stereo networks with public code available (just a comment, not an issue).
> >
> > - _Furthermore, as feature extraction is a very common component in deep-learning-based stereo matching networks, it is rather straightforward to integrate HODC into existing models, thus ensuring its universality._ I agree.
> >
> > **A5**
> >
> > Thanks for the clarification. I suggest the authors to include that in the paper.
> >
> > Best regards,
> >
> > Reviewer ApGF

---

> > > ### Author Response · Authors · 2024-08-08
> > >
> > > Dear Reviewer ApGF:
> > >
> > >
> > > Thanks for your response and insightful comment on our work.
> > >
> > > We are glad to see that your concerns have been addressed. As you suggested, we will include them in our final version. Thanks again for your constructive advice!
> > >
> > >
> > > Best regards,
> > >
> > > The Authors

---

### Official Review · Reviewer_uUe2 · 2024-07-11

**Soundness:** 4
**Presentation:** 4
**Contribution:** 3
**Rating:** 7
**Confidence:** 3

**Summary:**

The authors propose an additional training objective for image stereo matching methods in order to improve generalization from synthetic training data to real test images. It consists in a contrastive loss pushing image features aggregated according to some superpixels in one image to be similar to corresponding features aggregates in the other image (resp. dissimilar to non-corresponding feature aggregates), across different scales. The authors exploit object instance segmentation available in synthetic training data (SceneFlow dataset in this study) to perform this superpixel segmentation, and show in their experiments that the proposed objective function consistently brings performance improvements on various real-image datasets.

**Strengths:**

The paper is very well written and was a pleasure to read. The proposed idea is simple and well presented, and the results and ablations are convincing.

**Weaknesses:**

Minor remarks:
- the smooth L1 loss should be mathematically defined or a reference should at least be provided.
- Reporting baseline results without the proposed additionnal losses in Table 3 would make the table easier to interpret.

**Questions:**

- How do the different methods evaluated perform on Sceneflow test set? Reporting impact of the proposed approach on in-domain performance might be insightful.
- Ablation results in Table 3 using the proposed losses without object index map are rather good compared to the baseline methods without contrastive learning. This suggests that training for multi-scale correspondences may be of greater importance than using instance segmentation cues, which would downgrade the importance of semantics. Reporting results when using small base superpixel size (e.g. M=32), with contrastive losses but without object index priors could bring insights regarding this.

---

> ### Author Rebuttal · Authors · 2024-08-06
>
> Thanks for your positive feedback and helpful suggestions. We would like to make the following response to your questions:
>
> > Q1: the smooth L1 loss should be mathematically defined or a reference should at least be provided.
>
> A1: Thank you for pointing this out. We will add a reference to the smooth L1 loss in the final manuscript.
>
> > Q2: Reporting baseline results without the proposed additional losses in Table 3 would make the table easier to interpret.
>
> A2: Thanks for your advice. We will include the baseline results in Table 3 in the final version of the paper.
>
> > Q3: How do the different methods evaluated perform on Sceneflow test set? Reporting impact of the proposed approach on in-domain performance might be insightful.
>
> A3: Thank you for your suggestions. Following your suggestion, we have evaluated the in-domain performance of the models with and without our dual-level contrastive loss using the SceneFlow test set following our experiment settings. The results are listed in the table below.
>
> | Method                 | >1px    | >2px    | >3px    | EPE      |
> | ---------------------- | ------- | ------- | ------- | -------- |
> | PSMNet                 | 9.2     | 5.1     | 3.8     | 0.96     |
> | FC-PSMNet [1]          | 13.4    | 7.1     | 5.2     | 1.33     |
> | HVT-PSMNet [2]         | 9.2     | 5.2     | 3.9     | 1.04     |
> | **HODC-PSMNet (Ours)** | 9.0     | 5.0     | 3.8     | 1.03     |
> | GwcNet                 | 8.0     | 4.5     | 3.4     | **0.88** |
> | FC-GwcNet [1]          | 12.3    | 6.6     | 4.8     | 1.18     |
> | **HODC-GwcNet (Ours)** | **7.7** | **4.3** | **3.3** | 0.93     |
>
> Interestingly, the results revealed that our HODC can achieve high cross-domain performance as well as in-domain performance by establishing semantic and structural correspondence, with the same or even better threshold error rate compared to the baselines. Only a small decline is observed in the average end-point error metric.
>
> > Q4: Ablation results in Table 3 using the proposed losses without object index map are rather good compared to the baseline methods without contrastive learning. This suggests that training for multi-scale correspondences may be of greater importance than using instance segmentation cues, which would downgrade the importance of semantics. Reporting results when using small base superpixel size (e.g. M=32), with contrastive losses but without object index priors could bring insights regarding this.
>
> A4: Thanks for your advice regarding the role of instance segmentation cues. Instance segmentation cues allow us to generate regions with unambiguous semantic meanings, enabling us to establish semantically and structural correspondence accurately. Referring to Table 3, omitting the object prior will lead to a decline in the generalization performance of our HODC. We would like to highlight that $M$ denotes the segmentation scale, and a larger $M$ derives smaller base 'superpixel' sizes with finer representations. We restrict $M$ to be within $128$, as larger values will significantly increase computation cost and memory usage when calculating all pair correspondences.
>
>
>
> **References**
>
> [1] Revisiting Domain Generalized Stereo Matching Networks from a Feature Consistency Perspective. CVPR 2022.
>
> [2] Domain Generalized Stereo Matching via Hierarchical Visual Transformation. CVPR 2023.

---

> > ### Comment · Reviewer_uUe2 · 2024-08-13
> >
> > Thank you for this answer.
> > Regarding A4, indeed I was mislead by the 'scale' terminology and I confused $(N_w, N_h)$ with the segmentation size, which seems in fact $(W/N_w, H/N_h)$.

---

> > > ### Author Response · Authors · 2024-08-14
> > >
> > > We would like to thank you for your invaluable feedback on improving the quality of our paper. Kindly let us know if you might have further comments, and we will do our best to address them.

---

### Official Review · Reviewer_NWJx · 2024-07-12

**Soundness:** 3
**Presentation:** 3
**Contribution:** 3
**Rating:** 5
**Confidence:** 5

**Summary:**

This paper proposes a new framework for domain generalized stereo matching termed effective hierarchical object-aware dual-level contrastive learning (HODC). HODC improves the domain generalization ability of stereo matching by encouraging region-level information in extracted features. HODC can be easily integrated into current stereo matching architectures to improve their domain generalization ability.

**Strengths:**

1. The proposed method achieves SOTA domain generalization ability across various real-world stereo matching datasets.
2. The method is easy to implemented and suits various stereo matching architectures.
3. The paper is well-written.

**Weaknesses:**

1. The qualitative comparison in the article is insufficient. Considering that the generalization capability of stereo networks on public datasets is already quite good, the improvement of the method becomes more important from a visualization perspective. Especially since the article only uses PSMNet for qualitative comparison in Figure 1.
2. The paper lacks visualization analysis, making it difficult to discern how the proposed strategy impacts the learned feature representations.

**Questions:**

1. I would like to see this paper adds more qualitative visualizations, as well as evaluating the domain generalization performance on more challenging unseen domains such as the Booster and Spring datasets.

**Limitations:**

The authors have discussed the limitations.

---

> ### Author Rebuttal · Authors · 2024-08-06
>
> Thanks for your positive feedback and helpful suggestions. We would like to address your concerns as follows:
>
> > Q1: The qualitative comparison in the article is insufficient. Considering that the generalization capability of stereo networks on public datasets is already quite good, the improvement of the method becomes more important from a visualization perspective. Especially since the article only uses PSMNet for qualitative comparison in Figure 1.
>
> A1: Thanks for your advice. Due to space limitations, we included qualitative comparison results for PSMNet, GwcNet and IGEV on KITTI-2012, KITTI-2015, Middlebury, ETH3D in **Figures E**, **F**, and **G**, and results on the additional DrivingStereo [1] dataset in **Figure A** in the appendix of our paper.
>
> > Q2: The paper lacks visualization analysis, making it difficult to discern how the proposed strategy impacts the learned feature representations.
>
> A2: Thank you for expressing your concerns. Our proposed HODC aligns the regional features from the left image to the right under different scales, aiming at pulling the corresponding representations closer while pushing non-corresponding representations further apart. To validate the feasibility of this strategy in learning features with semantic and structural awareness, we visualized the representation similarity between a selected region within the left image and all pixels in the right image. As shown in **Figure 4**, the proposed HODC can accurately identify matched regions with limited ambiguities. More visualization results for feature representations on KITTI, Middlebury and ETH3D datasets are available in **Figures B**, **C** and **D** in the appendix of our paper.
>
> > Q3: I would like to see this paper adds more qualitative visualizations, as well as evaluating the domain generalization performance on more challenging unseen domains such as the Booster and Spring datasets.
>
> A3: Thanks for your suggestions. Following your advice, we evaluated the generalization performance of HODC on the challenging realistic Booster [2] dataset with quarter resolution and provided comparisons with other domain generalization methods. The results on the Booster training set are reported in the table below.
>
> | Method                 | >1px     | >2px     | >3px     |
> | ---------------------- | -------- | -------- | -------- |
> | PSMNet                 | 71.5     | 55.9     | 47.3     |
> | FC-PSMNet [3]          | 46.3     | 30.2     | 24.0     |
> | HVT-PSMNet [4]         | 37.0     | 24.6     | 19.2     |
> | **HODC-PSMNet (Ours)** | **36.0** | **23.0** | **18.0** |
> | GwcNet                 | 73.3     | 61.7     | 54.9     |
> | FC-GwcNet [3]          | 44.2     | 30.8     | 24.8     |
> | **HODC-GwcNet (Ours)** | 36.4     | 24.1     | 19.2     |
>
> The results on the Booster dataset indicate that HODC achieves satisfying results on challenging unseen domains, with significant improvements in generalization ability compared to baseline models. Following your advice, qualitative results on the Booster dataset are also provided in **Figure I** in the attached file in the global response.
>
> We would like to highlight that to further evaluate the generalization performance on challenging realistic scenarios, apart from using the widely used datasets, we have also provided quantitative and qualitative results on the DrivingStereo dataset, which contains diverse and challenging driving scenes. These results can be found in Appendix D of our paper.
>
>
>
> **References**
>
> [1] Drivingstereo: A large-scale dataset for stereo matching in autonomous driving scenarios. CVPR 2019.
>
> [2] Open challenges in deep stereo: the booster dataset. CVPR 2022.
>
> [3] Revisiting Domain Generalized Stereo Matching Networks from a Feature Consistency Perspective. CVPR 2022.
>
> [4] Domain Generalized Stereo Matching via Hierarchical Visual Transformation. CVPR 2023.

---

### Official Review · Reviewer_jWZr · 2024-07-13

**Soundness:** 3
**Presentation:** 3
**Contribution:** 3
**Rating:** 6
**Confidence:** 4

**Summary:**

This work proposed the hierarchical object-aware dual-level contrastive learning (HODC) framework for stereo matching. Their major technical contribution is a dual-level contrastive loss, which matches object features between intra- and inter-scale regions. Applying the proposed loss and only trained synthetic datasets, various networks achieve state-of-the-art performance across multiple realistic datasets.

**Strengths:**

-	The paper is well-written and easy to follow.
-	Instead of using the object information in a multi-task manner, the authors designed a contrastive loss with it, which is considered a novel idea.
-	The proposed loss can be easily plugged into network training.
-	The ablation study is thoroughly done.

**Weaknesses:**

As mentioned, it is not a new direction to explore the semantic and structural information in the stereo matching task (lines 43-45). Although the previous works took a different path when using this information, it is worth comparing the performance between those approaches and the method from this work.

**Questions:**

As mentioned in the Weaknesses section, I am curious about the performance of this work compared to the previous approaches that also utilize semantic information.

**Limitations:**

Limitations were discussed in the Appendix.

---

> ### Author Rebuttal · Authors · 2024-08-06
>
> Thanks for your positive feedback and helpful suggestions. We would like to make the following response to your questions:
>
> > Q1: As mentioned, it is not a new direction to explore the semantic and structural information in the stereo matching task (lines 43-45). Although the previous works took a different path when using this information, it is worth comparing the performance between those approaches and the method from this work.
>
> A1: Thanks for your suggestions. In our paper, we have highlighted that unlike our work, earlier works have explored semantic information by introducing subnetworks for semantic segmentation [1] or edge detection [2] within the stereo matching pipeline, adopting a direct multi-task methodology. Additionally, these prior works primarily focused on in-distribution scenarios instead of cross-domain ones.
>
> Following your recommendation for a more comprehensive analysis, we have conducted an experiment to compare the generalization performance of our HODC with the prior works [1, 2] that incorporate semantic information in stereo matching. As the source code for EdgeStereo [2] is unavailable, we replicated the experimental settings described in [2], utilizing only the Flyingthings3D training set for training, and then compared the generalization performance.
>
> The results presented in the table below demonstrate that networks trained with our HODC significantly outperform these previous approaches and exhibit superior generalization capability.
>
> | Method                 | KT15_EPE | KT15_3px | KT12_EPE | KT12_3px |
> | ---------------------- | -------- | -------- | -------- | -------- |
> | SegStereo [1]          | 2.2      | 11.2     | 2.1      | 12.8     |
> | EdgeStereo [2]         | 2.1      | 12.5     | 2.0      | 12.3     |
> | PSMNet                 | 6.4      | 29.9     | 5.5      | 27.3     |
> | **HODC-PSMNet (Ours)** | 1.4      | 6.3      | 1.2      | 6.0      |
> | **HODC-GwcNet (Ours)** | **1.2**  | **5.5**  | **0.9**  | **4.8**  |
>
>
>
> **References**
>
> [1] SegStereo: Exploiting Semantic Information for Disparity Estimation. ECCV 2018.
>
> [2] EdgeStereo: An Effective Multi-Task Learning Network for Stereo Matching and Edge Detection. IJCV 2020.

---

### Author Rebuttal · Authors · 2024-08-06

We thank all the reviewers for providing positive and insightful feedback. We are encouraged by the reviewers' appreciation that the paper is well-written, easy to follow and pleasure to read (Reviewer jWZr, NWJx, uUe2, ApGF), the ideas are novel yet easy to implement (Reviewer jWZr, NWJx), the results and ablations are convincing (Reviewer jWZr, uUe2, ApGF), the figures are clear and help the reader to understand the proposal and the results (Reviewer ApGF).

As suggested by the reviewers, we have now further conducted experiments to **1)** compare the generalization performance of HODC with prior works that incorporate semantic information for stereo matching, **2)** evaluate the in-domain performance of the models with and without our dual-level contrastive loss using SceneFlow test set and **3)** evaluate the generalization performance of our HODC on the challenging realistic Booster dataset [1]. We have also included additional qualitative comparisons on the Booster dataset, which can be found in the attached file. We hope that these results will provide additional clarification of our method to the reviewers.

**Reference**

[1] Open challenges in deep stereo: the booster dataset. CVPR 2022.

---

### Decision · Program_Chairs · 2024-09-25

**Decision:**

Accept (poster)

**Comment:**

The paper argued that existing stereo matching networks overlook the importance of extracting semantically and structurally meaningful features. It proposed an effective hierarchical object-aware dual-level contrastive learning (HODC) framework for domain generalized stereo matching. HODC can be integrated with existing stereo matching models in the training stage, requiring no modifications to the architecture. It was reviewed by four reviewers whereas the recommendations were 2XWeak Accept, 1XAccept and 1XBorderline accept. The authors provided rebuttal and there has been discussions. The reviewers are consistent with the paper's contributions. The authors are requested to make the revision.